# Neutralising SARS-CoV-2 RBD-specific antibodies persist for at least six months independently of symptoms in adults

Angelika Wagner[1], Angela Guzek[1], Johanna Ruff[1], Joanna Jasinska[1], Ute Scheikl[1], Ines Zwazl[1], Michael Kundi [2], Hannes Stockinger [3], Maria R. Farcet [4], Thomas R. Kreil[4], Eva Hoeltl[5] & Ursula Wiedermann [1✉]

## Abstract

**Background** In spring 2020, at the beginning of the severe acute respiratory syndrome coronavirus 2 (SARS-CoV-2) pandemic in Europe, we set up an assay system for large-scale testing of virus-specific and neutralising antibodies including their longevity.

**Methods** We analysed the sera of 1655 adult employees for SARS-CoV-2-specific antibodies using the S1 subunit of the spike protein of SARS-CoV-2. Sera containing S1-reactive antibodies were further evaluated for receptor-binding domain (RBD)- and nucleocapsid protein (NCP)-specific antibodies in relation to the neutralisation test (NT) results at three time points over six months.

**Results** We detect immunoglobulin G (IgG) and/or IgA antibodies reactive to the S1 protein in 10.15% ($n = 168$) of the participants. In total, 0.97% ($n = 16$) are positive for S1-IgG, 0.91% ($n = 15$) were S1-IgG- borderline and 8.28% ($n = 137$) exhibit only S1-IgA antibodies. Of the 168 S1-reactive sera, 8.33% ($n = 14$) have detectable RBD-specific antibodies and 6.55% ($n = 11$) NCP-specific antibodies. The latter correlates with NTs (kappa coefficient = 0.8660) but start to decline after 3 months. RBD-specific antibodies correlate most closely with the NT (kappa = 0.9448) and only these antibodies are stable for up to six months. All participants with virus-neutralising antibodies report symptoms, of which anosmia and/or dysgeusia correlate most closely with the detection of virus-neutralising antibodies.

**Conclusions** RBD-specific antibodies are most reliably detected post-infection, independent of the number/severity of symptoms, and correlate with neutralising antibodies at least for six months. They thus qualify best for large-scale seroepidemiological evaluation of both antibody reactivity and virus neutralisation.

## Plain language summary

Antibodies are proteins produced by the immune system in response to viruses. Antibodies against SARS-CoV-2, the virus that causes COVID-19, can be detected in the blood of people that have been previously infected. Here, we aimed to profile the levels of antibodies over 6 months in a group of 1655 Austrian adults working for the same company, with some working on-site and some working at home. Looking specifically at antibodies against the protein on the surface of the virus known as S1, we find that these are detectable in around approximately 10% of our group of adults and, of this group, 8% have antibodies against a specific part of the protein that binds its receptor on target cells. We observe that this specific subset of antibodies are most likely to persist up to 6 months, to be correlated with ability to neutralise the virus, and are associated with ongoing loss of taste and smell. These findings might have implications for monitoring of immunity, by helping us to understand which types of antibodies remain detectable and functional over time and how these relate to symptoms.

[1] Institute of Specific Prophylaxis and Tropical Medicine, Centre for Pathophysiology, Infectiology and Immunology, Medical University of Vienna, Vienna, Austria. [2] Centre for Public Health, Medical University of Vienna, Vienna, Austria. [3] Institute for Hygiene and Applied Immunology, Centre for Pathophysiology, Infectiology and Immunology, Medical University of Vienna, Vienna, Austria. [4] Global Pathogen Safety, Baxter AG, a Takeda Company, Vienna, Austria. [5] Health Centre Erste Bank, Erste Bank, Vienna, Austria. ✉email: ursula.wiedermann@meduniwien.ac.at

The ongoing severe acute respiratory syndrome coronavirus 2 (SARS-CoV-2) pandemic has led to dramatic restrictions in public life worldwide to mitigate the anticipated epidemic peak[1]. At the early stage of the pandemic, data were missing to estimate the actual number of infected individuals, including asymptomatic/oligosymptomatic cases that were left unreported due to the limitations in the polymerase chain reaction (PCR) testing capacity and strategy that was initially confined to the fulfilment of case definitions. It was, therefore, difficult to assess the actual risk of infection for employers with respect to shared workspaces and staff in contact with customers. Where possible, employers facilitated employees' work in home office mode with the intention to reduce the number of social contacts and consequently, the risk of infection[2].

In order to estimate past infections irrespective of symptoms and a preceding PCR test, specific and sensitive serological antibody tests, including neutralising assays, are necessary tools[3].

Several validated serological formats are now in use, based on enzyme-linked immunosorbent assays (ELISA) and chemiluminescence targeting different SARS-CoV-2 antigens[4]. The target antigens are the spike (S) protein with its receptor-binding domain (RBD), and the nucleocapsid protein (NCP). These antigens have been shown to induce robust antibody responses in infected individuals[5,6]. Another important question has emerged as to whether the detected antibodies also indicate protection against reinfection. In this regard, neutralising antibodies are likely to be considered as correlate of protection as they can lead to virus inactivation[6]. Along these lines, it is still unclear which test would prove most appropriate to describe transmission patterns and to determine immunity upon virus contact in overall, as well as in defined populations (in contrast to individual analysis/neutralisation tests [NTs]).

The goal of the current study was to analyse the seroprevalence of SARS-CoV-2-specific antibodies at the beginning of the pandemic and over several months in a representative cohort of employees from a large Austrian company. Therefore, several assays were included to identify the most accurate test for large-scale seroepidemiological analysis. The participants were included irrespective of a previous history of COVID-19 or experienced symptoms. The accumulated basic demographic data (gender, age, household size) and information on respiratory tract infections and symptoms, medical risk factors and travel history were analysed in context with the SARS-CoV-2-specific antibody test results. Employees with virus-reactive antibodies at the initial blood draw at the beginning of April 2020 were invited for follow-up blood draws at 3 and 6 months after study onset to analyse the persistence of the detected antibody levels. At 6 months, also participants without detectable virus-reactive antibodies at the initial blood draw were asked for a follow-up blood draw in order to detect seroconversion and to assess the development of seroprevalence in the overall study population over the last months.

Here, we show that RBD-specific antibodies are most reliably detected post-infection independently of the number/severity of symptoms and correlate with neutralising antibodies at least for 6 months.

## Methods

**Patients and samples.** We included 1655 serum samples of employees working for a large company in Vienna. While half of the staff continuously worked on-site with frequent client contacts, the other half worked from home at the beginning of this trial followed by a weekly rotation between home office and on-site work after the lockdown period in Austria. The blood samples were taken at the medical centre of the company between 2nd and 17th April 2020 and sent in for further analysis (to the

Institute of Specific Prophylaxis and Tropical Medicine at the Medical University of Vienna). Three months later, 156 of 168 participants with detectable S1-specific antibodies at the first blood draw came for a follow-up blood draw. Six months after the first blood draw 1292 of all 1655 participants participated in a follow-up blood draw, including 139 participants of those 168 that had detectable antibodies at the initial blood draw. The employees gave informed consent to SARS-CoV-2 serological testing and answered a questionnaire covering demographic data and their medical history, including current medications. Symptoms such as coughing, dyspnoea, thoracic pain, sore throat, rhinitis, elevated body temperature, fever, shivers, limb pain, weakness, headache, dysgeusia and/or anosmia, and gastrointestinal symptoms as described for COVID-19 were recorded, in addition to medical risk factors including those predisposing for a severe course of COVID-19. The risk factors were defined according to the Austrian Ministry of Social Affairs, Health, Care and Consumer Protection (chronic lung/respiratory disease, chronic cardiovascular disease, active cancer, immunosuppression and immunosuppressive drugs, chronic kidney disease, chronic liver disease with liver failure, diabetes, and arterial hypertension).

The ethics committee of the Medical University of Vienna approved this monocentric study (EK 1438/2020, EK 1746/2020).

**Testing for SARS-CoV-2-specific antibodies.** The SARS-CoV-2-specific antibody levels were measured with four different serological assays.

First, all sera were tested for SARS-CoV-2-specific immunoglobin (Ig) A and IgG antibodies using a commercial ELISA kit (Euroimmune®, Euroimmun Medizinische Labordiagnostika, Lübeck, Germany) according to the manufacturer's instructions. The antigen used in this semi-quantitative assay is the S1 domain of the SARS-CoV-2 S protein. The sera were diluted 1:101 before incubation. Results with a ratio below 0.8 were interpreted as negative, ratios between 0.8 and 1.1 as borderline and above 1.1 as positive, whereby ratios were calculated as optical density values of the control or patient sample divided by the optical density values of the calibrator. Samples within the borderline range and with ratios close to the cut-off of 0.8 or 1.1 (value from 0.7 to 1.2) were repeated in two independent tests and the geometric mean was used for the final result. Therefore, samples that were positive for S1-specific IgG, regardless of their S1-specific IgA result, were regarded as "IgG-positive", those with borderline values for S1-specific IgG after two repetitions, regardless of their IgA result, as "IgG-borderline" and those negative for S1-specific IgG but IgA-positive or IgA-borderline as "IgA-positive or -borderline" samples. No valid interpretation is currently possible in the case of isolated positive IgA findings. Samples negative for IgG and IgA were considered as "IgG- and IgA-negative".

Second, all positive and borderline samples were further evaluated for antibodies against the RBD of the S protein and NCP. The RBD-specific antibodies were determined using a commercial available ELISA for IgM and total antibodies (ab) (Beijing Wantai Biological Pharmacy Enterprise, Beijing, China) as described in the manufacturer's instructions, leading to borderline results when the ratio was between 0.9 and 1.1. Samples within the borderline range and with ratios close to the cut-off of 0.9 or 1.1 (i.e., between 0.8 and 1.2) were repeated in two independent tests and the geometric mean was used for the final result. NCP-specific IgG antibodies were tested by ELISA (Euroimmune®, Euroimmun Medizinische Labordiagnostika, Lübeck, Germany) applying the same criteria for calculating the results as for the S1 ELISA following the manufacturer's instructions. In addition, we measured total NCP-specific

antibodies in an automated sandwich electrochemiluminescence assay (Elecsys®, Roche Diagnostics, Mannheim, Germany; on a Cobas e 801). Results with a cut-off index of ≥1 were regarded as positive.

Finally, all IgG-positive sera and those with high IgA levels (Euroimmune ratio > 4) as well as a random sample ($n = 20$) of the seronegative sera were also tested/confirmed by a live virus-neutralising assay to evaluate the rate of functional antibodies. The SARS-CoV-2 NTs were done in cooperation with Takeda (Vienna, Austria).

**SARS-CoV-2 neutralisation assay**. SARS-CoV-2 neutralisation (NT) testing was done similar as previously described[7]. Briefly, Vero cells (ATCC CCL-81) sourced from the European Collection of Authenticated Cell Cultures (84113001) were cultured in TC-Vero medium supplemented with 5% fetal calf serum, L-glutamine (2 mM), nonessential amino acids (1x), sodium pyruvate (1 mM), gentamicin sulfate (100 mg/ml), and sodium bicarbonate (7.5%). SARS-CoV-2 strain BetaCoV/Germany/BavPat1/2020 was kindly provided by the Institute of Virology at Charité Universitätsmedizin, Berlin, Germany. For the SARS-CoV-2 neutralisation assays, samples were serially diluted 1:2 and incubated with 100 tissue culture infectious dose 50% ($TCID_{50}$) of SARS-CoV-2 per well. The samples were subsequently applied onto Vero cells seeded in tissue culture plates and incubated for 5 to 7 days, after which the cells were evaluated for the presence of a cytopathic effect and the SARS-CoV-2 neutralisation titre ($NT_{50}$), i.e., the reciprocal sample dilution resulting in 50% virus neutralisation, was determined using the Spearman-Kärber formula and reported as 1:X. Cut-off values varied between 1:3 and 1:7.7 $NT_{50}$ between the assay runs, depending on sample-predilution and level of sample cytotoxicity.

**Statistical evaluation**. For sample size considerations, we expected to compile data sets from ~800 employees continuing to work on-site with client contacts and some 700 home office workers. Based on the numbers of notified SARS-CoV-2 PCR-positive individuals in Vienna and an estimated number of 10% unreported cases, it was assumed that ~1% of the population had already had contact with the virus prior to the blood draw. Presuming that these figures would also apply to the employees included in the study, the effect size expressed as an odds ratio of 3 at a two-sided level of significance of 5% resulted in a statistical power of 77% to compare employees on-site and in home office (Fisher's exact probability test).

The data were evaluated for the two groups (working on-site and from home) and the subgroups stratified for age (15 to 25 years, 25 to 50 and above the age of 50). Seropositivity as a dependent variable was evaluated in a general linear model for binominal counts (see supplementary materials for more details). The primary predictor variable was defined according to the current workplace (home office or continuing to work with client contacts). Age, the number of household contacts, and the presence of underlying disease served as co-variables. To assess the relationship between various symptoms and seropositivity, a stepwise procedure was applied entering all mentioned covariates in one step and including symptoms in further steps with a significance level of 5% for inclusion and 10% for exclusion. The analyses were done using SPSS 26 (IBM Corp., New York, NY, USA) and the graphics were prepared with GraphPad Prism 7 (Graphpad Software, Inc., San Diego, CA, USA) or Excel (Microsoft Excel 2010, Microsoft Corporation, Redmond, WA, USA).

For changes in seroprevalence between the first blood draw and 6 months later, we calculated the ratio of new positives compared to new negatives applying the Chi² McNemar test and the difference in prevalences.

**Reporting summary**. Further information on research design is available in the Nature Research Reporting Summary linked to this article.

## Results

**Study populations, demographic data**. Overall, the study population consisted of 1655 volunteers, with an almost balanced female/male ratio (53.53%/46.47%) (Table 1 and Suppl. Fig. 1). The majority of the study population (62.30%) was in the medium-age group (25–50 years). Higher percentages of the medium-age and older age groups (53.83% and 62.78%, respectively) had been sent home to work in contrast to only 27.88% of the young age group. In relative terms, 52.33% of the total study population worked from home. About two-thirds (67.37%) of the volunteers were living in Vienna and one-third thus needed to commute to work. Almost one-third (27.98%) were sharing households with children younger than 15 years of age. Half of the participants (49.61%) had been travelling within the last 3 months before the first blood draw, either within Austria (winter/skiing holidays) or abroad (Suppl. Fig. 2). Regarding the use of public transport, 26.95% answered that they continued to use public transport during the lockdown period. Two-thirds (61.39%) maintained social contacts during the lockdown period and 2.18% stated that they had been in contact with SARS-CoV-2-infected individuals. In this regard, only two of these respondents were PCR-tested thereafter and both were PCR-negative. In total, 19 participants (1.15%) had previously been PCR-tested for SARS-CoV-2 before the first blood draw, of whom three showed a positive PCR result. Less than half of the participants (42.05%) reported that they had experienced symptoms compatible with COVID-19 within the last 3 months before the beginning of the study, with cough, rhinitis, sore throat, and fever being most frequently mentioned (Suppl. Fig. 3a). One quarter (23.63%) had risk factors in their medical history, which potentially predisposed for a severe COVID-19 disease (Suppl. Fig. 3b). However, 145 (8.76%) did not give details about the type of risk factors. Furthermore, 24.35% reported regular medication uptake (Table 1 and Suppl. Fig. 4).

**Sample analyses**. Serological testing for SARS-CoV-2 S1-specific IgG and IgA antibodies revealed that 1487 participants (89.85%) were seronegative (Fig. 1a). Thus, 10.15% displayed S1-reactive antibodies, among whom 16 (9.52%) were IgG-positive, 15 (8.93%) IgG-borderline, and 137 (81.55%) exhibited only IgA antibodies without IgG antibodies.

Further description of participants with antibodies reactive to S1 showed that COVID-19-associated symptoms were recorded in 39.88% of the participants with and in 42.17% without S1-reactive antibodies (Fig. 1b, c). With respect to the working situation, the highest rate of S1-positive or -borderline participants was home office workers before and during the beginning of the study (Fig. 1d). Regarding age, the highest number of participants with virus-reactive antibodies were between 25 and 50 years old (medium-age group) ($n = 111$, 6.71% of all participants) (Table 2).

The statistical analysis of the demographic and medical data revealed that three factors significantly correlated with S1 seropositivity (Table 3), i.e., (i) age, decreasing seropositivity with increasing age (age group 25–50: odds ratio (OR) 0.56; 0.32–0.99 95% confidence interval (CI) compared to age group >50: OR 0.38; 0.19–0.77 95% CI), (ii) home office (OR 1.91), and (iii) loss of taste and/or smell (OR 22.48; 7.75–65.22 95% CI). Concerning symptoms, anosmia/dysgeusia showed the highest probability for a positive virus (S1)-specific antibody result (OR

**Table 1 Demographic data by general characteristics and brief medical history.**

| Demographic data | | All participants (n = 1655) | | n | % | SEM |
|---|---|---|---|---|---|---|
| Gender | Female | | | 886 | 53.53% | (±1.23) |
| | Male | | | 769 | 46.46% | (±1.23) |
| Age group | Young | 15–25 a | Mean age = 22.40 (±0.15) | 226 | 13.66% | |
| | | | Home office | 63 | 27.88% | (±3.08) |
| | | | No home office | 146 | 64.60% | (±2.99) |
| | | | Not specified | 7 | 3.10% | (±1.16) |
| | Medium | 25–50 a | Mean age = 37.80 (±0.23) | 1031 | 62.30% | |
| | | | Home office | 555 | 53.83% | (±1.55) |
| | | | No home office | 409 | 39.67% | (±1.55) |
| | | | Not specified | 3 | 0.29% | (±0.17) |
| | Older | >50 a | Mean age = 54.75 (±0.18) | 395 | 23.87% | |
| | | | Home office | 248 | 62.78% | (±2.43) |
| | | | No home office | 123 | 31.13% | (±2.44) |
| | | | Not specified | 1 | 0.25% | (±0.25) |
| | | Not specified | | 3 | 0.18% | |
| Home office | No | | | 778 | 47.01% | (±1.23) |
| | Yes | | | 866 | 52.33% | (±1.23) |
| | Not Specified | | | 11 | 0.66% | (±0.20) |
| Residence | Vienna | | | 1115 | 67.37% | (±1.15) |
| | Outside Vienna | | | 531 | 32.08% | (±1.15) |
| | Not specified | | | 9 | 0.54% | (±0.18) |
| Children (≤15a) living in same houshold | No | | | 1184 | 71.54% | (±1.11) |
| | Yes | | | 463 | 27.98% | (±1.10) |
| | Not specified | | | 8 | 0.48% | (±0.17) |
| Travelling in the last 3 months | No | | | 806 | 48.76% | (±1.23) |
| | Yes | | | 822 | 49.61% | (±1.23) |
| | Not specified | | | 27 | 1.63% | (±0.31) |
| Use of public transport | No | | | 510 | 30.82% | (±1.14) |
| | Yes | | | 446 | 26.95% | (±1.10) |
| | Not specified | | | 699 | 42.24% | (±1.22) |
| Social contacts during lockdown | No | | | 104 | 6.28% | (±0.60) |
| | Yes | | | 1016 | 61.39% | (±1.20) |
| | Not specified | | | 535 | 32.33% | (±1.15) |
| Known contact with COVID-19 infected patients | No | | | 1617 | 97.70% | (±0.37) |
| | Yes | | | 36 | 2.17% | (±0.36) |
| | Not specified | | | 2 | 0.12% | (±0.09) |
| Previously tested for COVID-19 | No | | | 1630 | 98.49% | (±0.30) |
| | Yes | | | 19 | 1.15% | (±0.26) |
| | | | Positive | 3 | 15.79% | (±0.13) |
| | | | Negative | 5 | 26.32% | (±0.10) |
| | | | Not specified | 11 | 57.89% | (±0.20) |
| | Not specified | | | 6 | 0.36% | (±0.15) |
| Symptoms in the last 3 months | No | | | 956 | 57.76% | (±1.21) |
| | Yes | | | 696 | 42.05% | (±1.21) |
| | Not specified | | | 3 | 0,18% | (±0.10) |
| Personal risk factors / medical history | No | | | 1257 | 75.95% | (±0.88) |
| | Yes | | | 391 | 23.62% | (±0.87) |
| | Not specified | | | 7 | 0.42% | (±0.16) |
| Use of medication | No | | | 1246 | 75.29% | (±1.06) |
| | Yes | | | 403 | 24.35% | (±1.06) |
| | Not Specified | | | 6 | 0.36% | (±0.15) |

These data were analysed from the received questionnaires.

22.48), whereas the presence of any other reported symptom did not (OR 1.19; 0.80–1.77 95% CI).

Further characterisation of antibody responses in all 168 sera with detectable S1-specific antibodies revealed that 8.33% (14/ 168) had detectable RBD-specific total antibodies, whereas ten of these sera were also positive for S1-specific IgG, with two IgG borderlines and two IgG negatives (Fig. 2). The NCP-specific antibodies were positive in 6.55% (11/168) of the respondents with S1-reactive antibodies, with seven also being IgG-positive, one IgG-borderline and two IgG negatives. Of the S1-IgA-positive

or -borderline samples, two were positive for RBD and two for NCP (one of these for both).

SARS-CoV-2-specific neutralising antibodies, regarded as correlates of protective antibody responses showed that ten out of 16 S1-IgG-positive, two out of nine S1-IgG-borderline and only one of the S1-IgA-positive/borderline sera were also positive in the NT (Fig. 2 and Suppl. Fig. 5).

We then analysed whether the antibodies directed against S1, RBD or NCP, correlated with the presence of neutralising antibodies. Highest agreement with the NT was shown for

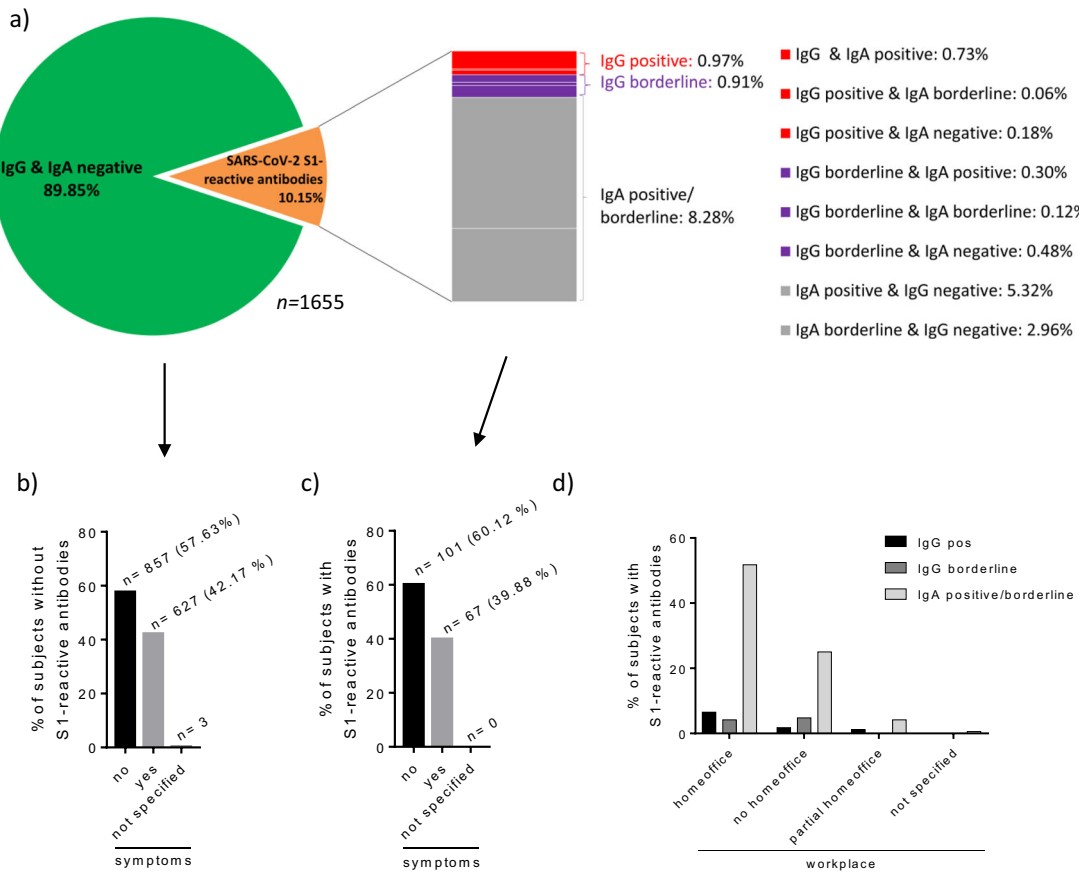

**Fig. 1 S1-reactive IgA and IgG antibody results.** Presented as proportions of the total study population ($n = 1655$) (**a**), of those without (**b**) and with detectable S1-reactive antibodies (**c**) dependent whether symptoms were recorded, according to workplace (**d**). Antibodies were measured in sera from the initial blood draw at day 0. Columns represent percentages of the total study population at the first blood draw (**b**–**d**).

---

**Table 2 S1-reactive IgA and IgG antibody results according to age group.**

| Age group | IgG & IgA negative | IgG positive | IgG borderline | IgA borderline/positive |
|---|---|---|---|---|
| 15–25 | 196 (86.73%) | 4 (1.77%) | 2 (0.88%) | 24 (10.62%) |
| 25–50 | 920 (89.23%) | 7 (0.68%) | 12 (1.16%) | 92 (8.92%) |
| >50 | 369 (93.42) | 5 (1.27%) | 1 (0.25%) | 20 (5.06%) |
| not specified | 2 | 0 | 0 | 1 |

Antibodies were measured in sera from the initial blood draw at day 0. S1 subunit of spike protein (S1).

---

**Table 3 Likelihood for seropositivity according to independent variables (predictors) such as home office, known contact to COVID-19 patients, children <15 years of age within the same household, residence in Vienna or outside, age group (comparison to the young 15–25 years old) and symptoms.**

| Predictor | Odds ratio | 95% confidence interval | p-value |
|---|---|---|---|
| Home office (yes/no) | 1.91 | 1.21–3.01 | 0.005 |
| Known contact to COVID-19 patients | 1.27 | 0.35–4.59 | 0.719 |
| Children < 15 years in the same household | 1.04 | 0.65–1.64 | 0.882 |
| Residence in Vienna (y/n) | 1.32 | 0.86–2.03 | 0.197 |
| Age group 15–24 | 1.00 | | |
| Age group 25–50 | 0.56 | 0.32–0.99 | 0.046 |
| Age group >50 | 0.38 | 0.19–0.77 | 0.007 |
| Any symptom | 1.19 | 0.80–1.77 | 0.397 |
| Anosmia/dysgeusia | 22.48 | 7.75–65.22 | <0.001 |

RBD-specific antibodies (total antibodies, kappa ($k$)=0.94; $p < 0.0001$ and IgM ($k = 0.89$; $p < 0.0001$) (Fig. 3)), indicating that these RBD-specific antibodies can be used as predictors for antibodies with virus neutralising properties. By contrast, S1-specific IgA failed to show any correlation with the neutralising antibody levels. Furthermore, the quantity (antibody level) of S1-specific IgG and IgA was not predictive for a positive NT result (Suppl. Fig. 5a, b).

All participants with neutralising antibodies reported symptoms, albeit of different qualities and quantities (Fig. 4a). The most prominent symptom recorded by 69.23% of the NT-positive participants was anosmia/dysgeusia (Table 4). However, the number of symptoms did not correlate with the level of the neutralisation titre (Fig. 4b). Of notice was that even in those with risk factors and neutralising antibodies, indicating past infection, neither a severe course of disease nor hospitalisation was reported. Regarding age distribution, the highest number of NT-positive test results was found in those aged 25 to 50 years, with slightly more males affected (53.85%) (Fig. 4b, d).

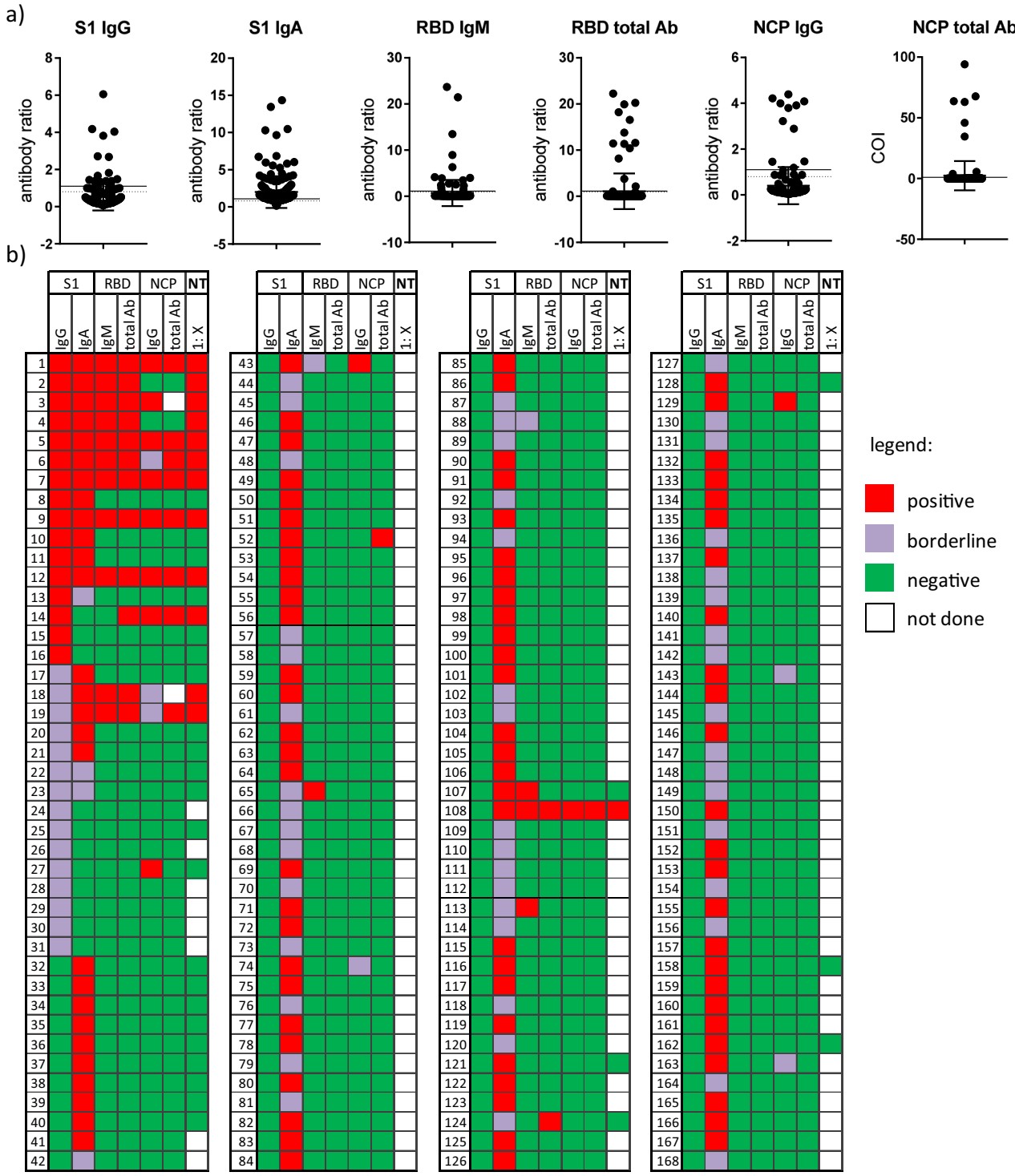

**Fig. 2 Individual antibody results.** Antibody results against different SARS-CoV-2 antigens expressed as antibody ratio or cut-off index (COI) (**a**) and as cumulative results (**b**) of all participants with S1-reactive antibodies (n = 168) at the initial blood draw (day 0). Dot plots shows mean + SD; **a**. Antibody results: red = reactive; violet = borderline; green = not reactive; white = n.d. (=not done). Antibody (Ab), S1 subunit of spike protein (S1), receptor-binding domain (RBD), nucleocapsid (NCP).

**Antibody persistence.** We were highly interested in exploring the longevity of seropositivity in our study population. Therefore, all participants with S1-reactive antibodies (n = 168) were invited for further blood draws after 3 and 6 months. Only the RBD-specific total antibody levels, which highly correlated with the neutralising antibodies, showed stable persistence, indicating that neutralising antibodies were maintained for at least 6 months

(Fig. 5). RBD-specific IgM was lost in 53.33% at already 3 months and remained relatively stable thereafter. In contrast, S1-specific IgG and IgA antibody levels rather tended to decline within 3 months in 37.5% and 53.68% of the participants, respectively (Fig. 5). Nevertheless, when S1-specific IgG antibodies were detectable at 3 months, they further persisted in 90% of the cases (9/10), also up to 6 months. NCP-specific IgG was lost in 9.09% at

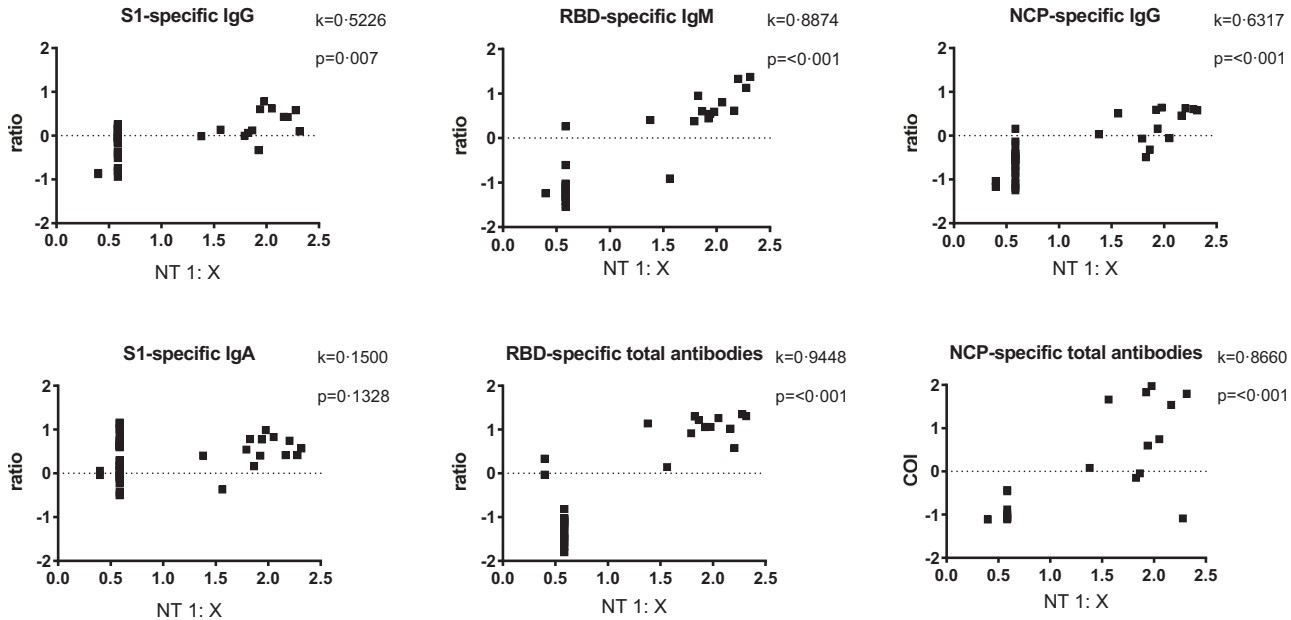

**Fig. 3 Correlation of SARS-CoV-2-specific antibody test results and the neutralisation test (NT).** Results from 41 participants with either detectable S1-specific IgG and/or IgA are expressed as antibody ratios or cut-off index (COI) with NT result expressed as titre 1:X measured in sera from the initial blood dray at day 0. Antibody ratios, COI and NT titres after logarithmic transformation. Kappa coefficient indicating agreement between the assays (very good > 0.8). The results presented as ratio or COI depend on the manufacturer´s instructions. Neutralisation test (NT).

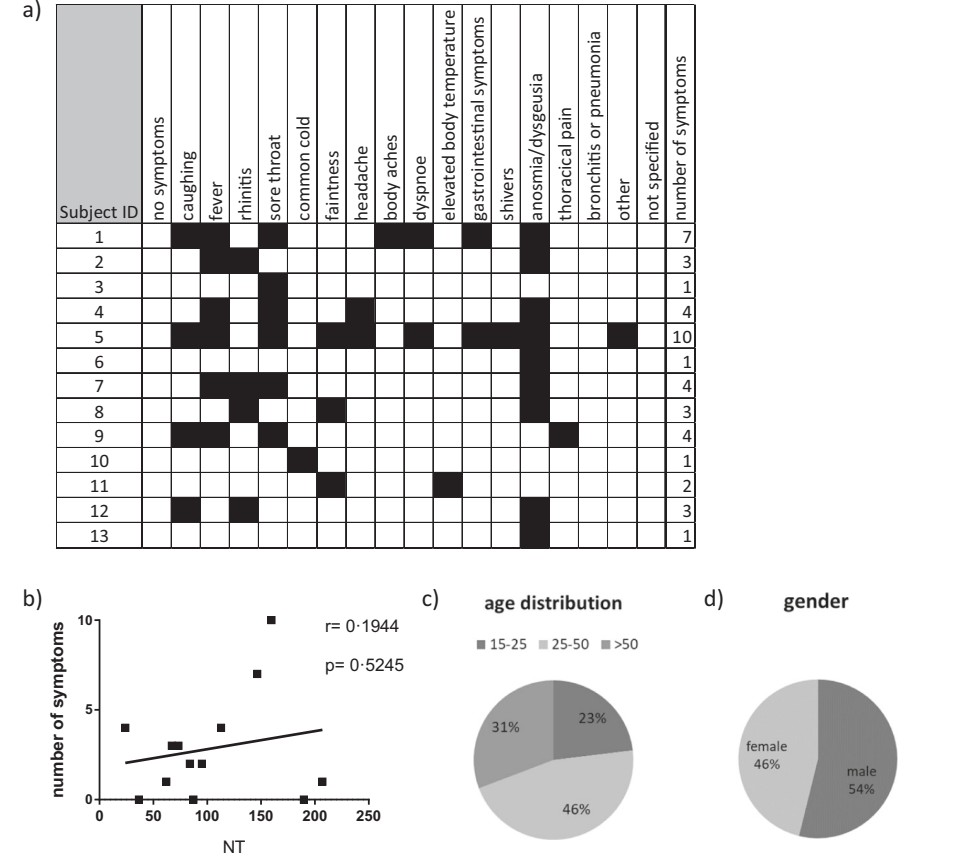

**Fig. 4 Symptoms, age and gender distribution of participants with neutralising antibodies.** Individual symptoms (**a**), correlation of the the number of symptoms with the neutralisation test (NT 1:X) result (**b**), distribution with regard to age group (**c**) and gender (**d**) among participants with neutralising antibodies evaluated in sera from the initial blood draw (day 0; $n = 41$). Neutralisation test (NT).

**Table 4 Overall prevalence of COVID-19 like symptoms in participants without and with neutralising antibodies in participants with neutralising antibodies evaluated in sera from the initial blood draw (day 0).**

| Symptoms | NT neg/ not done<br>n (%) | NT pos<br>n (%) | Prevalence | p-value |
|---|---|---|---|---|
| Cough | 341 (20.8%) | 4 (30.8%) | 1.2% | 0.489 |
| Fever/elevated body temp. | 214 (13.0%) | 7 (53.8%) | 3.2% | **0.001** |
| Sore throat | 205 (12.5%) | 6 (46.2%) | 2.8% | **0.003** |
| Rhinitis | 212 (12.9%) | 4 (30.8%) | 1.9% | 0.078 |
| Body aches | 55 (3.3%) | 1 (7.7%) | 1.8% | 0.362 |
| Faintness | 68 (4.1%) | 3 (23.1%) | 4.2% | **0.016** |
| Headache | 55 (3.3%) | 2 (15.4%) | 3.5% | 0.071 |
| Common cold | 87 (5.3%) | 1 (7.7%) | 1.1% | 0.510 |
| Shivers | 19 (1.2%) | 1 (7.7%) | 5.0% | 0.147 |
| Dyspnea | 27 (1.6%) | 2 (15.4%) | 6.9% | **0.021** |
| Thoracical pain | 14 (0.9%) | 1 (7.7%) | 6.7% | 0.112 |
| Gastrointestinal symptoms | 24 (1.5%) | 2 (15.4%) | 7.7% | **0.017** |
| Anosmia/dysgeusia | 6 (0.4%) | 9 (69.2%) | 60.0% | **<0.001** |
| Other | 23 (1.4%) | 1 (7.7%) | 4.2% | 0.174 |
| Any symptom | 669 (40.7%) | 13 (100.0%) | 1.9% | **<0.001** |

Neutralising test (NT).
Bold values refer to significant results (p-value < 0.05)

3 months, whereas 55.55% of NCP-specific IgG were lost at 6 months. Measurements of the total NCP-specific antibodies confirmed the result of the NCP-specific IgG assay and showed that the antibodies remained positive for 3 months (Fig. 5).

**Changes in seroprevalence over 6 months**. Six months after the first blood draw, we could evaluate antibody levels from 78.07% (1292/1655) of all participants, showing a non-significantly increased ratio of newly positives to those that became negative (Chi² McNemar $p = 0.212$) for S1-reactive IgG (Table 5). The prevalence of S1-reactive IgA antibodies decreased ($p > 0.001$) compared to the initial blood draw. The rate of RBD-specific IgM positivity as well as NCP IgG positivity increased over the 6-month period, although about 50% of previously positives became negative. With regard to the RBD-specific total antibodies, none of the previously positive antibodies declined to negative concentrations, while 2.1% of the previously negative became positive. This increase in RBD seropositivity as well as the stability of these antibodies over a period of 6 months is expressed as ratio >54 (Table 5).

**Discussion**

In our longitudinal study, we aimed to evaluate the seroprevalence and duration of antibody response against SARS-CoV-2 in a representative cohort of 1655 working adults over at least 6 months, whereby the onset of infection was unknown for the asymptomatic cases. An important aspect was to investigate which serological assay with respect to antigen specificity would be most appropriate for large-scale screening of past infections and seroprotection. Furthermore, we intended to investigate whether specific symptoms may serve as prediction markers of seropositivity and whether specific demographic parameters or working circumstances influenced the likelihood of virus contact/ infection. Finally, we also aimed to explore the duration of antibody responses and seroprotection up to 6 months.

Our study population comprised of three age groups ranging from 16 to 65 years representative of the Austrian adult working population. At the initial blood draw, antibody screening was performed using one of the first tests on the market, a S1-specific

ELISA. Of all participants, 10.15% had S1-specific IgG and/or IgA antibodies (above the cut-off, including borderline values) with 9.52% of these being positive for S1-specific IgG. These results correspond to those of a preceding Austrian study, in which 1544 random PCR samples were tested between the 1st and the 6th of April 2020, showing that the maximum prevalence of infected individuals was 0.33% (upper 95% confidence interval value)[8]. Our results of S1-specific IgG antibodies being prevalent in 0.97% of all participants would refer to the accumulated number of cases until that time. Similar rates have been reported for population-based seroprevalence studies in other European countries, while the rates have increased over time and varied substantially (0.4% to 14%) within different geographical regions relating to the occurrence of major infection clusters[9,10].

At the beginning of the pandemic and throughout the Austrian pre-lockdown period, the employer facilitated home office work for the majority of their employees, particularly those of higher age and with potential risk factors for severe COVID-19. Mainly the younger individuals (15 to 25 years), continued to work on-site during the overall study period with presumably the highest risk for exposure and indeed showing the highest levels of seropositivity. On the contrary, the percentage of S1-specific antibodies was lowest in the oldest age group, possibly explained by adequate compliance with the recommended hygiene measures and contact restrictions. Surprisingly, a higher prevalence of S1-specific antibodies was found in the group of participants working from home despite for age. However, this observation may in part be attributed to a bias related to the company's policy that all employees with respiratory symptoms should stay at home.

Evaluation of the recent medical histories revealed that 42% of all participants had experienced respiratory symptoms, such as cough, fever, sore throat, and rhinitis. Concurrent circulation of other respiratory infections during winter and early spring could explain the fact that only some of them had detectable antibodies against S1 (9.91%). Notably, while the overall appearance of symptoms did not correlate with seropositivity, anosmia, or dysgeusia can be regarded as predictive markers for infection with SARS-CoV-2 and subsequent seropositivity.

On the other hand, the relatively low number of S1 antibodies in relation to the recorded symptoms may be ascribed to a fast waning of this antibody type, as was recently described by other authors[11]. Furthermore, it was previously reported that ~10% of (non-hospitalised) patients presenting with mild COVID-19 did not mount detectable S1 antibody responses[12,13]. Thus, as we do not have PCR results from all our participants with recorded symptoms, some mild infections may have been missed in our study as well.

The improvement of the antibody test systems enabled detection of antibodies directed against different antigenic regions, such as the RBD and the NCP. Recent studies have indicated that the severity of disease has an impact on antibody levels and possibly also on their specificities[5,12–14]. Along these lines, we tested the antibody responses for these additional SARS-CoV-2 antigens within the group of participants, who had been identified as S1-seropositive. Only 15.5% of the sera from participants with S1-specific IgG antibodies also displayed antibodies directed against RBD or NCP. Notably, S1-specific IgA ratios, even at high levels, did not correspond at all with RBD- and NCP-specific antibody levels. This may be because the specificity of this test varies between 73%[15,16] and 94%, resulting in a very low estimated positive predictive value of ~39% according to Gereuts van Kessel et al.[17]. The low specificity of S1-specific IgA detection may be due to unspecific binding or cross-reactions of antibodies with other respiratory viruses[15,18]. Additionally, the role of SARS-CoV-2-specific IgA in sera of COVID-19 patients is not yet clear, but it may be

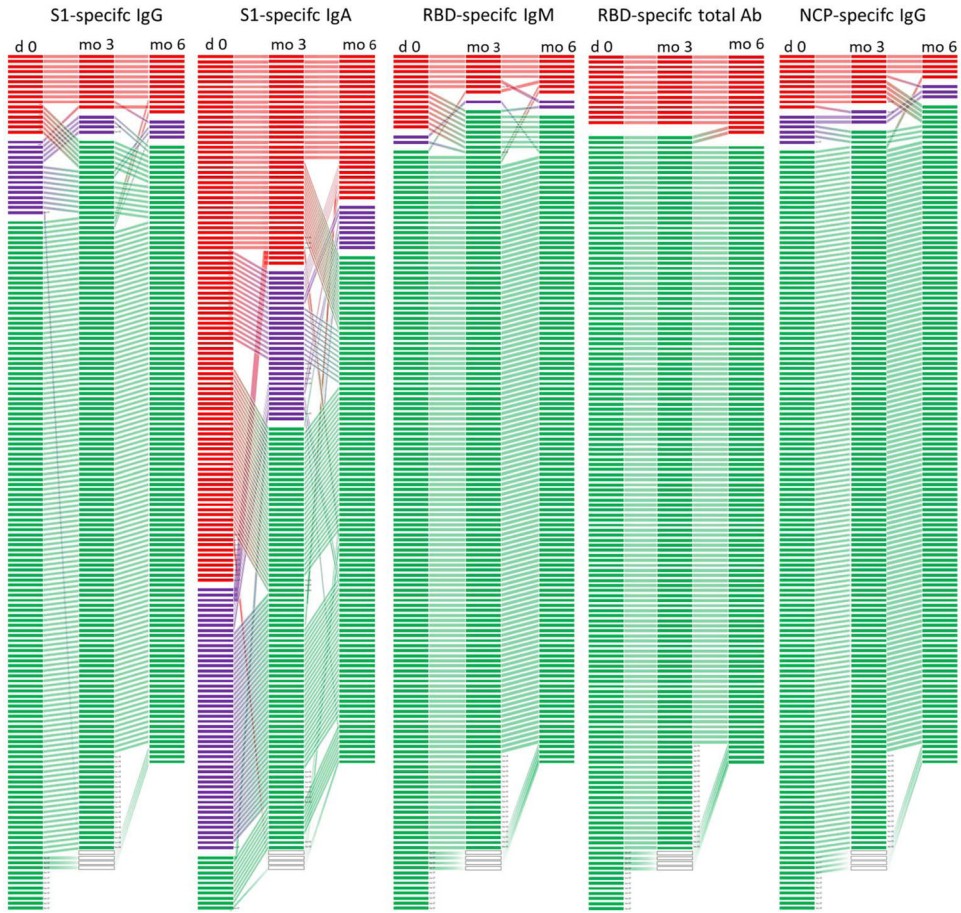

**Fig. 5 Changes in individual antibody test results over 6 months.** Development of individual antibody results at three different time points (day 0 ($n =$ 168), at three ($n = 152$) and 6 months ($n = 139$)) measured against different SARS-CoV-2 antigen specificities (S1, RBD and NCP) in the subgroup of those with detectable S1-reactive antibodies at the first blood draw. Each line represents one participant, red lines represents a positive result, violet a borderline result and negative results in green. S1 subunit of spike protein (S1), receptor-binding domain (RBD), nucleocapsid (NCP).

**Table 5 Changes in the seroprevalence between day 0 and 6 months.**

| Test | Ratio | *p*-value (against day 0) | Neg → pos (%neg) | Pos → neg (%pos) | Difference of prevalence (%) |
|---|---|---|---|---|---|
| S1-specific IgG | 1.56 | 0.212 | 2.0 | 55.2 | 0.7 |
| S1-specific IgA | 0.33 | <0.001 | 2.6 | 70.8 | −4.8 |
| RBD-specific IgM | 2.88 | 0.012 | 1.8 | 53.3 | 1.2 |
| RBD-specific total antibodies | >54 | <0.001 | 2.1 | 0.0 | 2.1 |
| NCP-specific IgG | 3.00 | 0.014 | 1.6 | 50.0 | 1.1 |

Changes are indicated by the ratio of newly positives to those that became negative; and the proportion of negatives that became positive (neg → pos) as well as the proportion of positives that became negative (pos → neg) and difference of prevalences of positives at month 6 and day 0; *p*-value assessed by Chi² McNemar. S1 subunit of spike protein (S1), receptor-binding domain (RBD), nucleocapsid (NCP).

involved in virus neutralisation in the early phase of COVID19 as suggested by Sterlin et al.[19]. The very high S1-specific IgA titres may still follow contact with the virus and could represent the mucosal activity of dimeric IgA—however, this needs to be further confirmed[13]. Thus, the overall lower specificity of both IgA and IgG antibodies against S1, as reviewed recently[20], also suggests that exclusive testing of S1-specific antibodies by ELISA is not optimal for seroepidemiological surveys in low-prevalence settings (<5%) due to the reduced positive predictive value for previous infection[4,21].

Importantly, we wanted to test whether the antibodies measured were also associated with neutralising capacity since they are regarded as surrogate marker of protection[22]. According to an experimental model with macaques, neutralising antibodies

against SARS-CoV-2 may play an essential role in protection against reinfection[23]. In our study, the total RBD antibodies showed the highest correlation (kappa = 0.9448) with the neutralising antibodies. Our data are supported by a recent study indicating that virus neutralisation is linked to B-cell epitopes of the S protein, in particular neutralising epitopes of its RBD[24,25]. In contrast to ~19% of S1-positive but asymptomatic participants, all of the participants with neutralising antibodies also showed symptoms— even if some had only one or two mild symptoms of which anosmia or dysgeusia was the most prevalent. Thus, anosmia or dysgeusia may even be regarded as a highly reliable diagnostic marker in very mild cases, as also proposed by other colleagues[26,27]. Of note, we could not find a correlation between the number of symptoms (implying severity) and the level of the

neutralisation titre as supported by other recently published data[28,29]. Our data further indicate that in individuals with anosmia/dysgeusia as a sole symptom, the quality and quantity of neutralising antibodies does not differ compared to those with several reported and typical COVID-19 symptoms (fever, cough, dyspnoea). Thus, our observation does not confirm suggestions that people with mild symptoms do not develop robust neutralising antibody responses[5,30].

Of major importance is the duration of the antibody responses. A recent study in COVID-19 patients showed that S1-specific and neutralising antibodies could last for up to 8 months[31,32]. In contrast, a study in healthcare workers, though in a rather small cohort, postulated that the virus-specific antibody responses to the S antigen are only of short duration, in particular in individuals with mild or asymptomatic courses of disease[33].

By analogy with the study authored by Patel et al.[33], the results of our investigation indicate that a high percentage of the S1-specific IgG and IgA antibodies declined already after 3 months of our observation period. Similarly, the NCP-specific antibodies also started to decline after 3 months. The drop in S1-specific antibodies was most evident and may be explained by an asymptomatic or oligosymptomatic course of the disease. Of interest, we here show that the kinetics of antibody decline differed according to antibody specificity over time. While the S1-specific antibodies decreased mostly within the first months, the NCP-specific IgG was solidly detectable for 3 months and thereafter declined until up to 6 months. In total contrast, the RBD-specific antibodies, correlating with neutralising antibodies, were consistently stable up to 6 months, and the number/severity of symptoms did not affect the duration of seroprotection. These results in mind, testing for RBD-specific antibodies should deliver the most reliable results to determine seroprevalence up to several months after infection. In contrast, cases may be lost already after 3 months when evaluating S1- or NCP-specific antibody response. Considering data from the previous SARS-CoV-1 pandemic, during which neutralising antibodies remained detectable in most patients for 2 years, it can be assumed that the neutralising antibodies will persist over the next months[34].

With the observation time of 6 months, we can also analyse changes in seroprevalence. Our results of the total study population show that in addition to the 0.85% RBD-positive participants at the initial blood draw, 2% additionally became positive after 6 months, most likely reflecting the recorded increase in cases in Austria during the autumn of 2020. However, we cannot exclude that some of those seroconversions at 6 months might be due to a delayed antibody response after infection, which has been described to occur up to weeks after symptom onset[35].

In summary, we report a low seroprevalence of 0.97% with regard to S1-specific IgG antibody levels within the adult population at the early beginning of the pandemic. However, regarding the detection of neutralising antibodies, the percentage is even lower with 0.85%. Importantly, these antibodies last for at least 6 months irrespective of a mild/oligosymptomatic or polysymptomatic course of the disease. Of all clinical symptoms, anosmia/dysgeusia is the most reliable symptom also associated with the generation of robust neutralising antibodies. In contrast, respiratory symptoms are not reliable diagnostic markers to predict antibodies or protection in the total study population. Large-scale seroprevalence studies can benefit from the use of a screening test with high SARS-CoV-2 specificity and RBD-specific assays could reliably detect SARS-CoV-2-specific antibodies for at least 6 months. Further evaluation covering up to 1 year is ongoing.

## Data availability
The datasets generated during and/or analysed during the current study are available from the corresponding author (UW also principal investigator) on reasonable request and after confirmation by the ethics committee. Data used to generate the charts and graphs in the main figures of the manuscript are provided in Supplementary Data 1.

## Code availability
No custom code or mathematical algorithm has been used in the manuscript.

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

## Acknowledgements
We would like to thank Sylvia Rudolf, Romana Hricova, Karin Baier, Maria Orola, Doaa al Mamoori, Tatjana Matschi, Vanessa Maurer, Barbara Schaar, Karin Schoiswohl, and Andrea Wendl for their excellent administrative and technical support. In addition, we would like to thank Thomas Perkmann (Laboratory Medicine, Medical University of Vienna) for the total NCP antibody measurements with the Roche Elecsys® assay. The contributions of Melanie Graf, Brigitte Kainz and Julius Segui (neutralisation assays) are gratefully acknowledged. SARS-CoV-2 was sourced via EVAg (supported by the European Community) and kindly provided by Christian Drosten and Victor Corman (Charité Universitätsmedizin, Institute of Virology, Berlin, Germany). This study received funding from the Austrian Ministry of Education, Science and Research within the research framework in relation to the coronavirus disease 2019 pandemic (GZ 2020 0225 104).

## Author contributions

Study conception and design: U.W., E.H., M.K. Development and methodology: A.W., A.G., J.J., U.W, H.S., M.R.F., T.R.K., M.K. Collection of the data: A.W., A.G., J.R., J.J., U.S., I.Z., M.R.F., T.R.K. Data analysis and interpretation: A.W., A.G., J.J., U.S., M.K., H.S., M.F., T.K., U.W. Writing all sections of the manuscript: A.W., A.G., U.W. Manuscript revision: A.W., A.G., J.R., J.J., U.S., I.Z., M.K., H.S., M.R.F., T.R.K., E.H., U.W.

## Competing interests
The authors A.W. A.G., J.R., J.J., I.Z., U.S., E.H., M.K., H.S. and U.W. declare no competing interest within the scope of this manuscript. M.R.F. and T.R.K. are employees of Baxter AG, now part of the Takeda group of companies, Vienna, Austria and have Takeda stock interest.
