## [Peer Review File · Communications Medicine]

Reviewers' comments:

Reviewer #1 (Remarks to the Author):

The paper presents several major criticisms

First, no information is given about the time from PSO and therefore the title and the type of persistence cannot be evidence-based. The authors should simply rewrite title and discussion explaining that they monitored at three and six months subjects using serological tests but without knowing the possible time of SARS-CoV-2 infection

there is a gross error in calculating the percentage of IgA positivity which is 8.2% and not 81.55% the disagreement between S1 antigen-based assay and RBD is impressive and should not be explained on the basis of current literature

Reviewer #2 (Remarks to the Author):

The COVID-19 pandemic has highlighted a need for large-scale immunological profiling and thorough understanding of virus-specific antibody responses in human population. Wagner et al performed a longitudinal analysis of antibody responses in a large cohort of 1,655 working adults that include individuals with unknown or suggested exposure history to SARS-CoV-2. They used a variety of semi-quantitative commercially available binding assays to assess IgA and IgG responses to known immunogenic antigens of SARS-CoV-2 that included S1, RBD, and NCP proteins. The authors performed neutralization assays using authentic virus with a limited number of serum samples that were pre-selected based on reactivity to S1. In addition, they analyzed various demographic parameters in an attempt to identify predictive markers of seropositivity in tested cohort. Their findings include (a), low seroprevalence (approx. 1%) of S1-specific IgG antibodies within the adult population early in the pandemic, (b), prevalence of S1-directed IgA responses for a cohort of the responders interpreted as positive from ELISA binding assay (c), RBD-specific binding positivity most accurately predicted antibody-mediated neutralization, and (d), no clear association between demographics measurements and positive responses. Overall, the results are of general interest to the field, and this study complements ongoing large scale serology studies (that assess antibody responses after seroconversion) by characterizing individual antibody responses to SARS CoV-2 when actual rate of infections was unknown. There are some comments below, which if addressed, would make the paper more accurate.

Specific comments:

Line 45, replace "protective neutralizing antibodies" with "neutralizing antibodies"

Line 47, replace "seroprevalence and seroprotection" with "antibody reactivity and neutralization"

Line 76, "gold standard" is a misused term. Please, replace with "correlate of protection".

Line 77-78, "However, such testing ...requires biosafety level 3 laboratory space...". Quantitative BSL2 neutralization assays including pseudotyped viruses and chimeric infectious viruses are now routinely used and have been shown to correlate well with BSL3 assays. The claim that BSL3 PRNT/CPE and similar assays are "gold standard" or "prevalent in the field" is no longer true.

Consider revising.

Lines 88 – 89 "...seroprotection and persistence of protective antibodies... "- antibody-mediated protection has not been evaluated in this study (e.g., passive polyclonal antibody transfer and viral challenge studies in animal models). The described assays measure antibody binding or antibody-mediated virus neutralizing activity.

Line 130 "...Results with a ratio below 0.8 were interpreted as negative..." - it is unclear which

qualitative/semi-quantitative parameters were assessed and used to calculate this ratio. The authors should explain what was exactly measured, how, and the output values that they used to define the ratio ((for example, ELISA optical density at 450 nm was measured and values minus background were compared to that from control wells; or timepoint 0 (first blood draw) measurement was compared to timepoint 2 (second blood draw) etc)).

Methods section, neutralization assay description did not indicate heat-inactivation of serum samples that removes the complement activity.

Line 223, it should be 8.93% instead of 893%

How do the authors explain that most individuals (82%) within the S1-reactive cohort exhibited only IgA antibodies without IgG antibodies? Could this be due to cross-reactivity of pre-existing IgA response to seasonal coronavirus infections (that has higher avidity when compared to IgG), so it could be unrelated to SARS-CoV-2 exposure? This hypothesis is supported by the author's data showing that serum from only one IgA-positive donor had neutralizing activity against SARS-CoV-2 in contrast to the IgG cohort. This should be discussed in more detail, as well as caveats for relying on S1-reactivity measurements alone to determine the history of SARS-CoV-2 exposure/demographics distribution, SARS-CoV-2 antibody responses persistence, etc.

Line 245, replace "surrogate of protection" with "correlate of protection".

Line 246, replace "functional antibodies" with "neutralizing antibodies" to not be confused with antibody Fc-mediated effector function assays.

Lines 246-247, "...regarded as the gold standard among the SARS-CoV-2-specific serological assays."

As noted above "gold standard" is a misused term; different BSL3 assays (e.g., PRNT, FRNT, endpoint CPE assay as in this study etc) are not standardized, and they are no longer prevalent in the field.

Please, revise to indicate the type of the virus used (authentic SARS-CoV2/strain) and the measurement (visual CPE assessment at the endpoint).

Line 253-255 – as indicated above, neutralizing activity is not equivalent to protection. Please, revise the conclusion to indicate that only binding to the RBD predicted neutralizing activity in serum.

Line 361 "Importantly, we wanted to test whether the antibodies measured were also associated with protection". As above, no protection studies were performed. Please, revise the text to replace "protection" with "neutralization".

Line 363 – 364 "...neutralising antibodies against SARS-CoV-2 may play an essential role in protection against reinfection^{23,24}" Ref. 23 reports NHPs protection in the context of both, neutralizing antibodies and adaptive CD8+/CD4+ T cell responses that are elicited after primary SARS-CoV-2 infection. It should be replaced with passive NHP sera polyclonal antibody transfer study by the same group showing solely antibody-mediated protection McMahan et al., Nature 2020 (<https://doi.org/10.1038/s41586-020-03041-6>).

Better references to replace Ref. 24 demonstrating NHP protection after passive transfer of clinical candidate monoclonal antibodies are:

Baum et al., Science 2020 (DOI: 10.1126/science.abe2402);

Zost et al., Nature 2020a (<https://doi.org/10.1038/s41586-020-2548-6>).

Line 366-368 "Our data are supported by a recent study indicating that virus neutralisation is linked to B cell epitopes of the S protein, in particular neutralizing epitopes of its RBD" – key references to support this statement are:

Robbiani et al., Nature 2020 (<https://doi.org/10.1038/s41586-020-2456-9>);

Zost et al., Nature Medicine 2020 (<https://doi.org/10.1038/s41591-020-0998-x>);

Liu et al., Nature 2020 (<https://doi.org/10.1038/s41586-020-2571-7>);

Rogers et al., Science 2020 (DOI: 10.1126/science.abc7520).

Figure 3 legend, a typo (“vexpressed”). Please, clarify representations for this figure in the legend (how many samples were used for this comparison and from which cohort)

One weakness of this study is the lack of quantitative ELISA binding measurements (semi-quantitative threshold-based assays were used), which limits data analysis.

Pavlo Gilchuk, Ph.D.
Sr. Staff Scientist
James E. Crowe Laboratory
Vanderbilt Vaccine Center

Reviewer #3 (Remarks to the Author):

Wagner et al. and colleagues studied the antibody response to SARS-COV-2 in a cohort of working adults in Austria. They measured antibodies against the S1 subunit of the spike, RBD, and N protein, as well as neutralizing antibodies. They found that 0.97% and 8.28% of 1655 adult employees were tested positive for anti-S1 IgG and IgA antibodies. Of them, about 8% were detected positive for anti-RBD and -N antibodies. They also found anti-RBD antibodies were stable and persistent for up to six months. Although the topic is interesting, several issues need to be addressed before publication.

Please re-organize the manuscript and make it more clear and logical, including abstract, methods, and results. In some instances, please remove no relevant information and focus on the main results.

Line 29, please explain the protective antibody. Is this a neutralizing antibody?

Please clearly describe how many adults you enrolled, how many serum samples were collected from the adults, and how many adults provided at least two serum samples in the main text. Except for quantitative measurement of neutralizing antibody, IgG and IgA antibodies were qualitatively measure.

Can you provide the cutoff value for neutralizing antibodies titer?

Line 232-236, can the author include OR and 95% in this section? It would be helpful for the reader to understand.

Can the author include the antibody type, such as IgG or IgA to RBD, S1, and NCP throughout the manuscript? The current version is really difficult to follow, especially for "sample analyses."

Can the author explain why some individuals had an increase of antibody titer six months compared to three months except for an increase in Austria cases?

Reviewers' comments: Point-by-point reply

Reviewer #1 (Remarks to the Author):

The paper presents several major criticisms

- 1) *First, no information is given about the time from PSO and therefore the title and the type of persistence cannot be evidence-based. The authors should simply rewrite title and discussion explaining that they monitored at three and six months subjects using serological tests but without knowing the possible time of SARS-CoV-2 infection*

We agree, in case of asymptomatic COVID19 cases where a PCR test was not performed, the exact time of the SARS-CoV-2 infection is not known. We adapted the title (referring to “at least” six months) and discussion that the timeline of six months refers to an observation period and does not refer to time of PSO (line 313, 407).

- 2) *there is a gross error in calculating the percentage of IgA positivity which is 8.2% and not 81.55%*

The 81.55% relate to the 137 participants with positive IgA levels within the 168 participants that were IgG and IgA seropositive (reactive) to S1 (line 233). Therefore no changes have been made.

- 3) *the disagreement between S1 antigen-based assay and RBD is impressive and should not be explained on the basis of current literature*

As described in the discussion part, our hypothesis for the disagreement between S1- and RBD-specific antibody responses is that in particular S1-specific IgA detection seems to be either unspecific or based on mucosal exposure to the virus without following infection, which is difficult to discriminate in this epidemiologic setting and would require further immunological evaluation (line 367 and following).

Reviewer #2 (Remarks to the Author):

The COVID-19 pandemic has highlighted a need for large-scale immunological profiling and thorough understanding of virus-specific antibody responses in human population. Wagner et al performed a longitudinal analysis of antibody responses in a large cohort of 1,655 working adults that include individuals with unknown or suggested exposure history to SARS-CoV-2. They used a variety of semi-quantitative commercially available binding assays to assess IgA and IgG responses to known immunogenic antigens of SARS-CoV-2 that included S1, RBD, and NCP proteins. The authors performed neutralization assays using authentic virus with a limited number of serum samples that were pre-selected based on reactivity to S1. In addition, they analyzed various demographic parameters in an attempt to identify predictive markers of seropositivity in tested cohort. Their findings include (a), low seroprevalence (approx. 1%) of S1-specific IgG antibodies within the adult population early in the pandemic, (b), prevalence of S1-directed IgA responses for a cohort of the responders interpreted as positive from ELISA binding assay (c), RBD-specific binding positivity most accurately predicted antibody-mediated neutralization, and (d), no clear association between demographics measurements and positive responses. Overall, the results are of general

interest to the field, and this study complements ongoing large scale serology studies (that assess antibody responses after seroconversion) by characterizing individual antibody responses to SARS CoV-2 when actual rate of infections was unknown. There are some comments below, which if addressed, would make the paper more accurate.

Specific comments:

- 1) *Line 45, replace “protective neutralizing antibodies” with “neutralizing antibodies”*

This has been revised accordingly, since a correlate of protection has not been established yet (line 46).

- 2) *Line 47, replace “seroprevalence and seroprotection” with “antibody reactivity and neutralization”*

We have replaced this wording (line 48).

- 3) *Line 76, “gold standard” is a misused term. Please, replace with “correlate of protection”.*

We now rephrased that neutralising antibodies are likely to be considered as correlate of protection (line 77-78).

- 4) *Line 77-78, “However, such testing ...requires biosafety level 3 laboratory space...”. Quantitative BSL2 neutralization assays including pseudotyped viruses and chimeric infectious viruses are now routinely used and have been shown to correlate well with BSL3 assays. The claim that BSL3 PRNT/CPE and similar assays are “gold standard” or “prevalent in the field” is no longer true. Consider revising.*

Taking into account the development of pseudotyped virus assay to determine neutralising antibodies and we have adapted the text by excluding the argument that testing for neutralising antibodies needs BSL3 and is laborious and time-consuming (line 78-80).

- 5) *Lines 88 – 89 “...seroprotection and persistence of protective antibodies... “- antibody-mediated protection has not been evaluated in this study (e.g., passive polyclonal antibody transfer and viral challenge studies in animal models). The described assays measure antibody binding or antibody-mediated virus neutralizing activity.*

This sentence has been corrected accordingly and the sentence on testing for seroprotection has been now excluded (line 90-91).

- 6) *Line 130 “...Results with a ratio below 0.8 were interpreted as negative...” - it is unclear which qualitative/semi-quantitative parameters were assessed and used to calculate this ratio. The authors should explain what was exactly measured, how, and the output values that they used to define the ratio ((for example, ELISA optical density at 450 nm was measured and values minus background were compared to that from control wells; or timepoint 0 (first blood draw) measurement was compared to timepoint 2 (second blood draw) etc)).*

The assay detecting S1-specific antibodies was done according to the manufacturer's instruction. Ratios were calculated as optical density values measured at 450nm of the control or patient sample divided by the optical density values of the calibrator as indicated in the manufacturer's instructions allowing a semiquantitative estimation of the result. We now included these details on how we calculated the ratio into the method section (line 137-139).

- 7) *Methods section, neutralization assay description did not indicate heat-inactivation of serum samples that removes the complement activity.*

The NT assays were performed according to established and previously published protocols (Schwaiger et al; <https://www.ncbi.nlm.nih.gov/pubmed/32941626>). When establishing the SARS-CoV-2 neutralisation test the effect of complement inactivation was tested, showing no effect on the results. Therefore samples were not heat inactivated before being tested.

- 8) *Line 223, it should be 8.93% instead of 893%*

We now corrected this number (line 232).

- 9) *How do the authors explain that most individuals (82%) within the S1-reactive cohort exhibited only IgA antibodies without IgG antibodies? Could this be due to cross-reactivity of pre-existing IgA response to seasonal coronavirus infections (that has higher avidity when compared to IgG), so it could be unrelated to SARS-CoV-2 exposure? This hypothesis is supported by the author's data showing that serum from only one IgA-positive donor had neutralizing activity against SARS-CoV-2 in contrast to the IgG cohort. This should be discussed in more detail, as well as caveats for relying on S1-reactivity measurements alone to determine the history of SARS-CoV-2 exposure/demographics distribution, SARS-CoV-2 antibody responses persistence, etc.*

In the discussion (line 367-380) we pointed out, that the discrepancy in S1-specific IgA and IgG values may be due to unspecific binding or cross-reactivity with other respiratory viruses including other coronaviruses. However, we did not further explore the cross-reactivity of antibodies specific for other coronaviruses from the same samples as this would be beyond the focus of our current study. Another possibility to explain the discrepancy of S1-specific IgA and IgG results may be that low exposure to SARS-COV-2 at mucosal surfaces may result in preferential IgA production when further infection is abrogated. Since S1-specific IgA binding does not correlate well with S1-specific IgG and other antibody specificities to RBD and NCP we also stated that diagnosis of SARS-CoV-2 infection should not be exclusively based on S1-specific IgA detection.

- 10) *Line 245, replace "surrogate of protection" with "correlate of protection".*

In accordance with the introduction (comment regarding line 76) we now state that neutralising antibodies most likely correlate with protection (257-260).

- 11) *Line 246, replace "functional antibodies" with "neutralizing antibodies" to not be confused with antibody Fc-mediated effector function assays.*

This has been revised accordingly rewriting the sentence also with regard to comment 10 (line 257-260).

- 12) Lines 246-247, "...regarded as the gold standard among the SARS-CoV-2-specific serological assays." As noted above "gold standard" is a misused term; different BSL3 assays (e.g., PRNT, FRNT, endpoint CPE assay as in this study etc) are not standardized, and they are no longer prevalent in the field. Please, revise to indicate the type of the virus used (authentic SARS-CoV2/strain) and the measurement (visual CPE assessment at the endpoint).

The strain used (BetaCoV/Germany/BavPat1/2020) is described in the methods section (line 170). The wording related to the „gold standard“ has been revised by rewriting the whole sentence (see also comment 10 and 11; line 257-260).

- 13) Line 253-255 – as indicated above, neutralizing activity is not equivalent to protection. Please, revise the conclusion to indicate that only binding to the RBD predicted neutralizing activity in serum.

We now adapted the respective wording (line 265-269).

- 14) Line 361 "Importantly, we wanted to test whether the antibodies measured were also associated with protection". As above, no protection studies were performed. Please, revise the text to replace "protection" with "neutralization".

This statement has been revised accordingly (line 382).

- 15) Line 363 – 364 "...neutralising antibodies against SARS-CoV-2 may play an essential role in protection against reinfection^{23,24}" Ref. 23 reports NHPs protection in the context of both, neutralizing antibodies and adaptive CD8+/CD4+ T cell responses that are elicited after primary SARS-CoV-2 infection. It should be replaced with passive NHP sera polyclonal antibody transfer study by the same group showing solely antibody-mediated protection McMahan et al., Nature 2020 (<https://doi.org/10.1038/s41586-020-03041-6>).

Better references to replace Ref. 24 demonstrating NHP protection after passive transfer of clinical candidate monoclonal antibodies are:

Baum et al., Science 2020 (DOI: 10.1126/science.abe2402);

Zost et al., Nature 2020a (<https://doi.org/10.1038/s41586-020-2548-6>).

Thank you for the suggestion we replaced the references for those more adequate to underline our statement (line 384).

- 16) Line 366-368 "Our data are supported by a recent study indicating that virus neutralisation is linked to B cell epitopes of the S protein, in particular neutralizing epitopes of its RBD" – key references to support this statement are:
Robbiani et al., Nature 2020 (<https://doi.org/10.1038/s41586-020-2456-9>);
Zost et al., Nature Medicine 2020 (<https://doi.org/10.1038/s41591-020-0998-x>);
Liu et al., Nature 2020 (<https://doi.org/10.1038/s41586-020-2571-7>);
Rogers et al, Science 2020 (DOI: 10.1126/science.abc7520).

We now updated the references with regard to our statement (line 388).

17) *Figure 3 legend, a typo (“vexpressed”). Please, clarify representations for this figure in the legend (how many samples were used for this comparison and from which cohort)*

We corrected the typo and included the information that samples from 41 participants with detectable S1-specific IgG and/or IgA were included into the analysis (line 599-600).

18) *One weakness of this study is the lack of quantitative ELISA binding measurements (semi-quantitative threshold-based assays were used), which limits data analysis.*

At the time we performed our study quantitative SARS-CoV-2 specific ELISAs were not commercially available. We agree that the quantitative titer levels would give additional information especially with regard to the persistence of the antibody response.

However, in this current study our goal was to define which antibody assay and which antibody-specificity is most appropriate to be analysed in seroepidemiological studies – here we describe that the RBD-specific antibodies are a robust correlate for neutralising antibodies.

Reviewer #3 (Remarks to the Author):

Wagner et al. and colleagues studied the antibody response to SARS-COV-2 in a cohort of working adults in Austria. They measured antibodies against the S1 subunit of the spike, RBD, and N protein, as well as neutralizing antibodies. They found that 0.97% and 8.28% of 1655 adult employees were tested positive for anti-S1 IgG and IgA antibodies. Of them, about 8% were detected positive for anti-RBD and -N antibodies. They also found anti-RBD antibodies were stable and persistent for up to six months. Although the topic is interesting, several issues need to be addressed before publication.

1) *Please re-organize the manuscript and make it more clear and logical, including abstract, methods, and results. In some instances, please remove no relevant information and focus on the main results.*

According to the reviewers comments we removed less relevant information throughout the manuscript which might have distracted from the focus of the manuscript.

2) *Line 29, please explain the protective antibody. Is this a neutralizing antibody?*

We were referring to neutralising antibodies and have now adapted the wording accordingly (line 29). Neutralising antibodies, however, will likely be regarded as a correlate of protection.

3) *Please clearly describe how many adults you enrolled, how many serum samples were collected from the adults, and how many adults provided at least two serum samples in the main text.*

The requested information has now been added to the manuscript (methods section line 113-116)).

- 4) *Except for quantitative measurement of neutralizing antibody, IgG and IgA antibodies were qualitatively measure.*

At the time we performed our study only semi-quantitative ELISA were commercially available. We agree that the quantitative titer levels would give additional information especially with regard to the persistence of the antibody response. However, in this current study our goal was to define which antibody assay and which antibody-specificity is most appropriate to analyse in seroepidemiological studies— here we describe that the RBD specific antibodies are a robust correlate for neutralising antibodies.

- 5) *Can you provide the cutoff value for neutralizing antibodies titer?*

Cut-off values varied between 1:3 and 1:7.7 NT₅₀ between the assay runs, depending on sample-predilution and level of sample cytotoxicity (now included in line 177-178).

- 6) *Line 232-236, can the author include OR and 95% in this section? It would be helpful for the reader to understand.*

We now included the information on OR and 95% CI into the requested paragraph (line 243-248).

- 7) *Can the author include the antibody type, such as IgG or IgA to RBD, S1, and NCP throughout the manuscript? The current version is really difficult to follow, especially for "sample analyses."*

Throughout the revised manuscript we have now consequently described the antibody specificity throughout the manuscript and hope that thereby the text has improved in clarity.

- 8) *Can the author explain why some individuals had an increase of antibody titer six months compared to three months except for an increase in Austria cases?*

A inter-individual difference in the kinetic of antibody production resulting in a delayed antibody response in some individuals could be one explanation of this observation apart from the overall increase in incidence – we now discussed this possibility in the discussion section accordingly (line 427).

REVIEWERS' COMMENTS:

Reviewer #1 (Remarks to the Author):

The authors made great efforts in answering to the criticisms raised by the referees. However, the manuscript is very long and there result section contains information that should be anticipated in the "material and Method" section. This should make more easy to read and understand the content of the paper

Reviewer #2 (Remarks to the Author):

The authors addressed all my comments in a satisfactory manner.

Pavlo Gilchuk, Ph.D.
Sr. Staff Scientist
Vanderbilt Vaccine Center

Reviewer #3 (Remarks to the Author):

The author responded well to my comments.